# LEARNING PSEUDO 3D REPRESENTATION FOR EGO-CENTRIC 2D MULTIPLE OBJECT TRACKING

## ABSTRACT

Data association is a knotty problem for 2D Multiple Object Tracking due to the object occlusion. However, in 3D space, data association is not so hard. Only with a 3D Kalman Filter, the online object tracker can associate the detections from LiDAR. In this paper, we rethink the data association in 2D MOT and utilize the 3D object representation to separate each object in the feature space. Unlike the existing depth-based MOT methods, the 3D object representation can be jointly learned with the object association module. Besides, the object's 3D representation is learned from the video and supervised by the 2D tracking labels without additional manual annotations from LiDAR or pretrained depth estimator. With 3D object representation learning from Pseudo 3D (P3D) object labels in monocular videos, we propose a new 2D MOT paradigm, called P3DTrack. Extensive experiments show the effectiveness of our method. We achieve state-of-the-art performance on the ego-centric datasets, KITTI and Waymo Open Dataset (WOD). Code will be released.

## 1 INTRODUCTION

Multiple Object Tracking (MOT) is the core component of the perception system for many applications, such as autonomous driving and video surveillance. In the deep learning era, metric learning helps the network to learn better object affinity between frames for object association in MOT (Wojke et al., 2017; Brasó & Leal-Taixé, 2020; He et al., 2021). Another hot trend is jointly learning object detection and association, termed as end-to-end MOT (Meinhardt et al., 2022; Zeng et al., 2022; Zhou et al., 2022). In these methods, they use the shared query to generate the object's bounding box in each frame belonging to the same track. This kind of design makes the neural network jointly learn object representation and data association across frames. Previous attempts demonstrate that precise association is crucial in MOT. However, in 2D MOT, object association remains a significant challenge due to object occlusion. The presence of many partially visible objects in congested scenarios like shopping malls and traffic jams makes incorrect association nearly impossible to prevent. Several approaches have aided the data association module with complex appearance models and image-space motion models to address the challenging 2D data association. Although these techniques have proven effective, they do not target the main problem of object association, that is, trying to associate 3D objects in 2D image space.

Conversely, in 3D MOT, many works demonstrate that object association is nearly a trivial problem, even with a simple motion model. ImmortalTracker (Wang et al., 2021b), in particular, reveals that using the 3D Kalman Filter to model motion from the LiDAR 3D bounding boxes, the wrong association only occurs *once* in the entire Waymo Open Dataset (WOD) dataset. This significant gap between 2D and 3D MOT reveals that association in a higher-dimensional space is much simpler than in a low-dimensional space. Therefore, inspired by this observation, this paper aims to address the 2D object association problem in a 3D space.

Recent works (Khurana et al., 2021; Dendorfer et al., 2022) explore the most straightforward way to lift 2D association to 3D space, that is utilizing the off-the-shelf depth model. However, such methods are not effective enough for three reasons. (1) It is hard to estimate the temporal consistent scene-level depth from the monocular images. (2) The camera's intrinsic parameters are different, so the pretrained depth estimation model has limited generalization ability in the tracking scenes. (3) Association with explicit depth is sub-optimal since the depth estimation and the association part

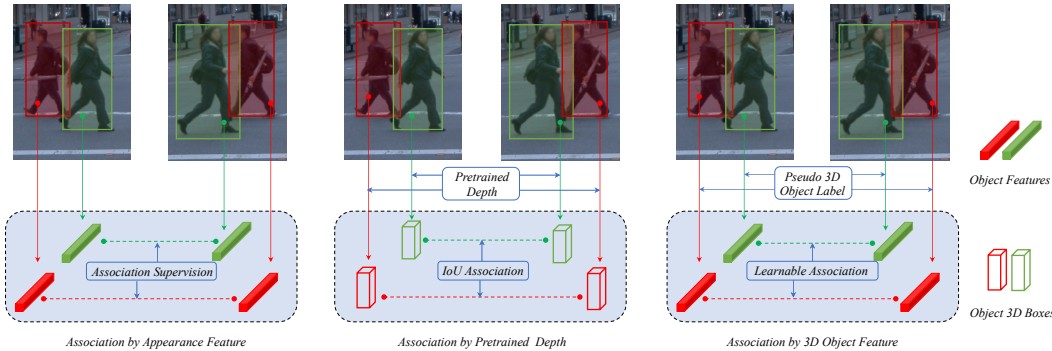

Figure 1: **Three paradigms for 2D MOT.** Left: Association by appearance feature learning. Middle: Association by lifted 3D objects from the pretrained depth model. Right: Association by learnable 3D representation from pseudo 3D object labels.

are isolated without joint optimization. Meanwhile, without joint optimization, the association is also sensitive to the noise of explicit depth.

Distinct from these works, we want to learn the representation containing 3D position information for the *objects*, and the object's 3D representation can be jointly learned with the association module, as shown in Fig. 1. Besides, due to the expensive cost and the additional sensors (e.g., LiDAR) to obtain the 3D annotations, we want to dig the 3D object representation only from 2D tracklet labels annotated in the videos without any other ground truth 3D information. In this paper, we propose a new video-based 3D representation learning framework and the 2D Multiple Object Tracking algorithm with the 3D object representation, called P3DTrack. P3DTrack mainly consists of three parts: (1) pseudo 3D object label generation, (2) jointly learning framework including 3D object representation and object association learning, (3) a simple yet effective online tracker with learned 3D object representation. As for pseudo 3D object label generation, inspired by Structure-from-Motion (Schönberger & Frahm, 2016), the scene can be reconstructed in 3D space from the camera motion. The 3D object is located as a part of the reconstructed scene. Fortunately, with the 2D bounding boxes in video frames, we can find the 3D object's reconstructed point clouds lie in the intersection area of the object frustums of multiple frames. By finding the main cluster of the reconstructed object point clouds, the 3D object position can be annotated as the center of the cluster. After that, the pseudo label offers P3DTrack supervision to learn the 3D representation of objects. The representations are then fed into a graph-matching-based object association module for joint optimization. Then the tracker can track the objects frame-by-frame with the robustness of heavy occlusion and similar appearance. In summary, our work has the following contributions:

- We propose a new online 2D MOT paradigm, called P3DTrack. P3DTrack utilizes the jointly learned 3D object representation and object association.

- We design the 3D object representation learning module, called 3DRL. With 3DRL, the object's 3D representation is learned from the video and supervised by the 2D tracking labels without any additional annotations from LiDAR or pretrained depth estimator.

- The extensive experiments on the large-scale ego-centric datasets, KITTI and Waymo Open Dataset (WOD) show that we achieve new state-of-the-art performance.

## 2 RELATED WORK

### 2.1 MULTIPLE OBJECT TRACKING

Multiple Object Tracking (MOT) aims to associate the same object in a video. To track the object, linear motion models, like Kalman Filter (KF) (Kalman, 1960), and appearance models (Wojke et al., 2017) from re-identification are the core components in the tracker. However, the 2D MOT task suffers from object occlusion, camera shake, and similar appearance. Many methods tend to use graph neural networks (Brasó & Leal-Taixé, 2020; Weng et al., 2020b; He et al., 2021) and attention

mechanism (Zeng et al., 2022; Meinhardt et al., 2022; Zhou et al., 2022) to aggregate the object features across intra- and inter-frames. Some researchers reveal that learning object features and object association jointly (Xu et al., 2020; He et al., 2021) can help to obtain more discriminative object features. Transformer-based end-to-end MOT (Zeng et al., 2022; Meinhardt et al., 2022) build a new paradigm with track query to represent a whole track. Score-based detection selection, such as ArTIST (Saleh et al., 2021) and ByteTrack (Zhang et al., 2022b), also helps the tracker to keep high-quality detections and tracks. In 3D MOT, 3D Kalman Filter is a very common practice. AB3DMOT (Weng et al., 2020a) propose a simple KF-based online tracker. Recent work (Wang et al., 2021b) has revealed that with only 3D KF, LiDAR-based 3D MOT is almost association error-free evaluated on the mainstream autonomous driving datasets.

Recently, with the development of monocular 3D object detection (Park et al., 2021; Zhang et al., 2021b; Wang et al., 2021c) and multi-camera 3D object detection (Wang et al., 2022b; Huang et al., 2021; Li et al., 2022b), camera-based 3D multiple object tracking is emerging. In the early years, Mono3DT (Hu et al., 2019) is the first to learn monocular 3D object detection and 3D multiple object tracking. MUTR3D (Zhang et al., 2022a) extends the transformer-based tracker to camera-based 3D MOT. PF-Track (Pang et al., 2023) is the new state-of-the-art camera-based 3D MOT method utilizing past and future trajectory reasoning. Although these methods are a little similar to ours on the 3D representation learning level, we focus on different key problems. We care more about whether the hidden 3D information in the video can help the association in 2D MOT, so we do not use the manually annotated 3D object labels.

## 2.2 3D REPRESENTATION FROM VIDEO

In this subsection, we visit the existing work representing objects in the 3D space, especially for learning from videos. Generally speaking, recovering the 3D representation from a video can be divided into scene reconstruction and representing objects in 3D space. With the help of multi-view geometry, Structure from Motion (SfM) (Schönberger & Frahm, 2016) is a practical algorithm to estimate the depth of each keypoint and recover the sparse 3D structure of the static scene from a movable camera. Multi-view Stereo (MVS) (Schönberger et al., 2016) is the dense scene reconstruction method from every pixel. In robotics, Simultaneous Localization and Mapping (SLAM) (Davison, 2003; Newcombe et al., 2011; Mur-Artal et al., 2015) utilizes a continuous video clip to estimate the robot's ego-pose and construct a sparse or dense map for planning and controlling.

In the perception field, a common practice is representing 3D objects with cuboid bounding boxes, defined as the 3D object detection task (Shi et al., 2020; Fan et al., 2022). However, most vision-based 3D object detection methods (Park et al., 2021; Zhang et al., 2021b) leverage the neural network to fit the cuboid labels from *LiDAR* annotations. As for 3D video object detection, inspired by scene reconstruction, DfM (Wang et al., 2022a) and BEVStereo (Li et al., 2022a) refine the object depth from the video. BA-Det (He et al., 2023) is the first to represent the consistent 3D object explicitly in the video via object-centric bundle adjustment.

Another kind of 3D representation from the video is to learn video consistent depth. SfM-Learner (Zhou et al., 2017) is a pioneer work taking view synthesis as supervision. (Luo et al., 2020; Zhang et al., 2021c) are the methods of learning consistent video depth and smooth scene flow with test-time training. These methods learn from traditional Structure-from-Motion and treat temporal warping consistency as geometric constraints. However, our method learns object 3D representation rather than the dense scene-level depth.

## 3 METHODOLOGY

In this section, we introduce the new 2D MOT paradigm with 3D object representation. The pipeline contains three parts: (1) a CenterNet (Zhou et al., 2019)-based 2D object detection branch; (2) a joint learning framework to learn 3D object representation and object association; (3) an online 2D tracker with 3D object representation in the inference stage, called P3DTrack.

The 2D object detection branch is mainly from CenterNet (Zhou et al., 2019), with DLA-34 (Yu et al., 2018) backbone fusing multi-scale features. The output feature map of the backbone is with stride 4. The detection heads include the center heatmap prediction head and the bounding box regression head. Following FCOS (Tian et al., 2019), the regression target is $(l^*, t^*, r^*, b^*)$, corre-

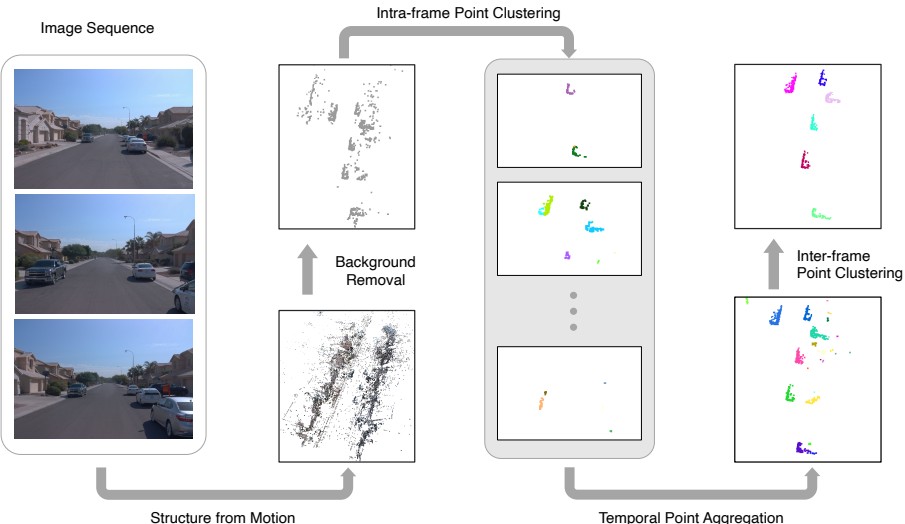

Figure 2: **Illustration of pseudo 3D object label generation process.** After reconstructing the scene with Structure-from-Motion, intra-frame and inter-frame point clustering are adopted. The pseudo 3D object label can be defined as the barycenter of the cluster.

sponding to the distance from the regression center to the sides of the bounding box. In the inference stage, non-maximum suppression (NMS) is performed as a post-process, different from the original CenterNet.

## 3.1 GENERATING PSEUDO 3D OBJECT LABELS

The existing works, which either utilize the pretrained depth estimator supervised by LiDAR or learn the object representation with human-annotated 3D labels, are not pure 2D MOT paradigms. We want to learn the 3D object representation only from the monocular video. In this section, as shown in Fig. 2 and Alg. 1 in Appendix, we introduce how to generate the pseudo 3D object labels from the video clips taken from the moving cameras based on the Structure-from-Motion algorithm.

SfM (Schönberger & Frahm, 2016) system can be applied to obtain the 3D position of each keypoint $\mathbf{P}_i$ in the global frame by solving the bundle adjustment problem. Thus the scene can be reconstructed as a set of 3D points. Note that not all scenes can be reconstructed well. When the camera movement is relatively small, the reconstruction performance is bad. So, we filter the videos with a low speed of ego-motion. Besides, in the traditional SfM system, the static scene can be well-reconstructed but the moving objects can not be reconstructed because their movement is not the same as the ego-motion and are filtered in the outlier filtering step by RANSAC (Fischler & Bolles, 1981). Thanks to the generalization capability of the deep neural network, these inherent problems of traditional SfM are easily solved. Please refer to Sec. 3.2 for more details.

After reconstructing the 3D points in the global frame, we filter the points that can be projected to the foreground regions in the images, and then we perform the Intra-frame Point Clustering (Intra-PC) to select the main connected component of the 3D keypoints for each object $\mathbf{B}_j^t$. The distance threshold of a cluster is $\delta$. That means we only consider the biggest cluster belonging to the object, which can filter the noise points in the background regions or caused by keypoint mismatching.

Besides, we cluster the 3D keypoints for the second stage, called Inter-frame Point Clustering (Inter-PC), to further filter the noise points in the global frame for all objects together: In each cluster, the number of points must exceed the threshold $\kappa$. In general, the cluster number is less than the number of tracks, so some tracks will not be assigned to the 3D object labels. In 3D object representation learning, these objects are only supervised by 2D representation labels and be ignored in 3D.

After clustering the 3D points, we should match the cluster with the maximum probability corresponding to the tracklet. We define the matching strategy that if the cluster can be only projected

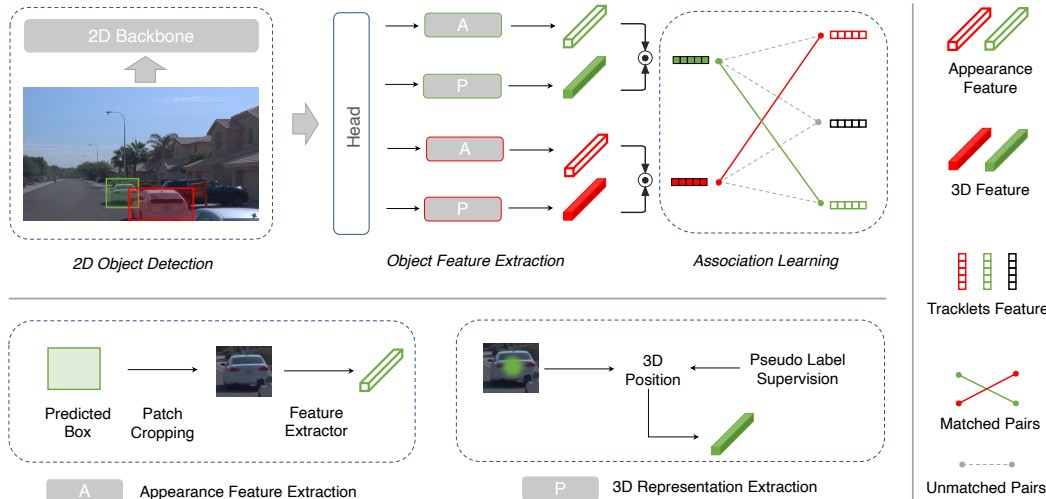

Figure 3: **Pipeline of P3DTrack.** The object detector is based on CenterNet. We learn the 3D object representation and data association module jointly from the pseudo 3D object labels.

into *one* bounding box $\mathbf{B}_j$ with id $j$, we match the cluster to the tracklet $j$. If not, we assign the cluster to the tracklet whose 2D bounding box can involve the maximum number of the reprojected points. Note that we only label static objects and this matching strategy guarantees the same 3D object position in the global frame for a tracklet, so the 3D object representation learning can keep *consistent* in a tracklet and the association can be learned well.

To supervise the 2D bounding box with 3D representation, we need to define the target 3D location for each ground truth 2D bounding box. Two kinds of definitions are common in practice. The first is the barycenter, which is the average 3D position of the cluster in the global frame. The second definition is the minimum circumscribed cuboid for the cluster. In the experiments, we find that with these two definitions, tracking performance is nearly the same. So, we finally choose the barycenter $\mathbf{o}_j^{t*} = [x_j^{t*}, y_j^{t*}, z_j^{t*}]^\top$ in the camera frame as the 3D position target for the object. In summary, we obtain the initial 3D representation with a cluster of 3D points for each tracklet. Then we need to learn the generalized 3D representation for each object and utilize the 3D object representation to associate the objects between frames.

### 3.2 3D REPRESENTATION AND ASSOCIATION LEARNING

**3D object representation learning.** In this subsection, we introduce the proposed 3D object representation learning module, called the 3DRL, shown in Fig. 3. 3DRL module is an additional head of the object detection model CenterNet to represent 3D attributes of the objects, with a shared DLA-34 backbone. We denote the output of the backbone feature as $\mathbf{F}_t \in \mathbb{R}^{H \times W \times C}$. Like FairMOT (Zhang et al., 2021a), we choose the 2D object center position to extract the 3D feature $^{(3D)}\mathbf{f}_j^t$ of the object $\mathbf{B}_j^t$, which will be further utilized for object tracking. In the following, we will explain how to extract the 3D object feature. The 3D branch first uses MLPs as the feature encoder to encode the backbone features, and the 3D object feature is

$$\mathbf{o}_j^t = \texttt{MLP}(\mathbf{F}_t)[c_j(u), c_j(v)], \tag{1}$$

where $[c_j(u), c_j(v)]$ is the position of 2D object center on the feature map $\mathbf{F}_t$, and $\mathbf{o}_j^t = [x_j^t, y_j^t, z_j^t]^\top$ is the 3D position of the object center in the camera frame. Learned from the generated pseudo 3D object labels, we define the 3D object position auxiliary loss with uncertainty following (Kendall & Gal, 2017; Zhang et al., 2021b),

$$\mathcal{L}_{3D} = \frac{||\mathbf{o}_j^t - \mathbf{o}_j^{t*}||_1}{\sigma^2} + \log(\sigma^2), \tag{2}$$

where $\sigma^2$ is the learnable 3D position uncertainty for the object $j$.

Then the 3D object representation is predicted by an additional fully-connected layer

$$^{(3D)}\mathbf{f}_j^t = \mathtt{fc}(\mathbf{o}_j^t).\tag{3}$$

Note that even though only the position of the static object is supervised, the network can easily make the 3D position prediction of the moving object, because the network does not distinguish the object movement in one frame. So, whether the object is moving does not affect 3D object representation learning.

**Appearance model.** Besides the 3D object representation, following the mainstream 2D MOT methods, we use an object re-identification network as the appearance model:

$$^{(2D)}\mathbf{f}_j^t = \mathtt{ReID}(\mathbf{I}^t, \mathbf{B}_j^t).\tag{4}$$

The architecture of the appearance model is the ResNet50 backbone followed by a global average pooling layer and a fully connected layer with 512 channels. Triplet loss (Hermans et al., 2017) is used for robust appearance feature learning. The appearance model is trained on WOD training set for 20 epochs and the feature is also used for training the object association module, but will not be updated.

**Object association.** The proposed object association model contains two parts: a GNN-based feature aggregation module and a differentiable matching module. Given the object's 3D representation and appearance feature, we define the object feature for

$$^{(0)}\mathbf{f}_j^t = [^{(2D)}\mathbf{f}_j^t, {}^{(3D)}\mathbf{f}_j^t],\tag{5}$$

where $[\cdot]$ is the concatenation operation and 3D representation and appearance features are $L_2$-normalized first. We adopt an $L$-layer cross-frame GNN to aggregate the object features from different frames:

$$^{(l+1)}\mathbf{f}_j^t = \mathtt{MLP}\left(^{(l)}\mathbf{f}_j^t + \frac{||^{(l)}\mathbf{f}_j^t||_2\,^{(l)}\mathbf{m}_j^{t-1}}{||^{(l)}\mathbf{m}_j^{t-1}||_2}\right), l \in [0, L-1],\tag{6}$$

where the aggregated message $\mathbf{m}_j^{t-1}$ is calculated from

$$^{(l)}\mathbf{m}_j^{t-1} = \sum_{j'=1}^{n_{t-1}} {}^{(l)}w_{j,j'}\,^{(l)}\mathbf{f}_{j'}^{t-1}.\tag{7}$$

Averaging weight $w_{j,j'}$ is cosine similarity between the normalized features

$$^{(l)}w_{j,j'} = \cos(^{(l)}\mathbf{f}_j^t, {}^{(l)}\mathbf{f}_{j'}^{t-1}).\tag{8}$$

The matching layer is defined as a differentiable Quadratic Problem (He et al., 2021), and the association loss can be the binary cross-entropy loss between the matching results and the matching ground truth.

### 3.3 Tracking with 3D Object Representation

After obtaining the 3D object representation and training the object association model, we propose our online tracker P3DTrack. In this section, we explain how the tracker infers frame by frame. Similar to DeepSORT (Wojke et al., 2017), we match detections with existing tracklets in each frame. The tracklet's appearance feature is the average of appearance features in each frame. As for 3D object representation, we utilize a 3D Kalman Filter to model the 3D object motion of the tracklet. The 3D representation of the tracklet is from the predicted 3D position using 3D KF and encoded with Eq. 3. During the object association stage, we use the association model in Sec. 3.2. Besides, we balance the 3D representation and appearance similarity with weight $\alpha$ and $1 - \alpha$, respectively.

## 4 Experiments

### 4.1 Dataset and Evaluation Metrics

We conduct experiments on mainstream ego-centric MOT datasets, KITTI and Waymo Open Dataset. Please note that other datasets, such as MOT17 and MOT20, include numerous surveillance videos without camera motion. These datasets are not suitable for evaluating our method.

| Method | Backbone | MOTA ↑ | IDF1 ↑ | FN ↓ | FP ↓ | ID Sw. ↓ | MT ↑ | ML ↓ |
|---|---|---|---|---|---|---|---|---|
| IoU baseline (Lu et al., 2020) | ResNet-50 | 38.25 | - | - | - | - | - | - |
| Tracktor++ (Bergmann et al., 2019) | ResNet-50 | 42.62 | - | - | - | - | - | - |
| RetinaTrack (Lu et al., 2020) | ResNet-50 | 44.92 | - | - | - | - | - | - |
| QDTrack (Fischer et al., 2022) | ResNet-50 | 55.6 | 66.2 | 514548 | 214998 | 24309 | 17595 | 5559 |
| P3DTrack (Ours) | DLA-34 | 55.9 | 65.6 | 638390 | 74802 | 37478 | 13643 | 8583 |

Table 1: Results on WOD tracking val set using py-motmetrics library.

| Method | +Label | +Data | HOTA | AssA | ID Sw. | DetA | AssRe | AssPr | LocA | MOTA |
|---|---|---|---|---|---|---|---|---|---|---|
| QD-3DT (Hu et al., 2022) | 3D GT | | 72.77 | 72.19 | 206 | 85.48 | 74.87 | 89.21 | 87.16 | 85.94 |
| Mono3DT (Hu et al., 2019) | 3D GT | | 73.16 | 74.18 | 379 | 85.28 | 77.18 | 87.77 | 86.88 | 84.28 |
| OC-SORT (Cao et al., 2023) | | PD | 76.54 | 76.39 | 250 | 86.36 | 80.33 | 87.17 | 87.01 | 90.28 |
| PermaTrack (Tokmakov et al., 2021) | | PD | 78.03 | 78.41 | 258 | 86.54 | 81.14 | 89.49 | 87.10 | 91.33 |
| RAM (Tokmakov et al., 2022) | | PD | 79.53 | 80.94 | 210 | 86.33 | 84.21 | 88.77 | 87.15 | 91.61 |
| QDTrack (Fischer et al., 2022) | | | 68.45 | 65.49 | 313 | 85.37 | 68.28 | 88.53 | 86.50 | 84.93 |
| TrackMPNN (Rangesh et al., 2021) | | | 72.30 | 70.63 | 481 | 83.11 | 73.58 | 87.14 | 86.14 | 87.33 |
| CenterTrack (Zhou et al., 2020) | | | 73.02 | 71.20 | 254 | 84.56 | 73.84 | 89.00 | 86.52 | 88.83 |
| LGM (Wang et al., 2021a) | | | 73.14 | 72.31 | 448 | 82.16 | 76.38 | 84.74 | 85.85 | 87.60 |
| DEFT (Chaabane et al., 2021) | | | 74.23 | 73.79 | 344 | 83.97 | 78.30 | 85.19 | 86.14 | 88.38 |
| P3DTrack (Ours) | | | **74.59** | **76.86** | **219** | 83.21 | 80.66 | 86.67 | 86.28 | 85.60 |

Table 2: Tracking results on KITTI *test* set.

**Waymo Open Dataset (WOD).** We conduct our experiments mainly on the large-scale autonomous driving dataset, Waymo Open Dataset (WOD). WOD has 798 sequences for training, 202 for validation, and 150 for testing. More than 1 million images are taken from 5 surrounding cameras. Such a big dataset is suitable for more general feature learning in MOT. The official evaluation metrics are CLEAR MOT metrics (Bernardin & Stiefelhagen, 2008). MOTA is the main metric to evaluate the number of false positives, false negatives, and mismatches. The number of mismatches is the most important to evaluate the data association performance. Besides, to evaluate more detailed data association performance, we also report IDF1 metric (Ristani et al., 2016) using py-motmetrics[1]. The main results are reported using all 5 cameras on WOD val set. The ablation study is conducted with the FRONT camera on WOD val set.

**KITTI dataset.** We conduct some additional experiments on KITTI (Geiger et al., 2012) dataset. KITTI MOT dataset has 21 sequences for training and 29 sequences for testing. The official metrics are mainly based on HOTA (Luiten et al., 2021), including the detection metrics DetA, LocA, and the association metric AssA. KITTI also evaluates the CLEAR MOT (Bernardin & Stiefelhagen, 2008) metrics.

## 4.2 IMPLEMENTATION DETAILS

**Training.** We use a DLA-34 (Yu et al., 2018) as the backbone without any neck, and the head is with 2 layers of 3*3 convolutions and MLP. The resolution of the input images is 1920*1280. If the input size is smaller than it, we will use zero padding to complete the image. Our implementation is based on PyTorch (Paszke et al., 2019) framework. We train our model on 8 NVIDIA A40 GPUs. Adam (Kingma & Ba, 2014) optimizer is applied with $\beta_1 = 0.9$ and $\beta_2 = 0.999$. The learning rate is $8 \times 10^{-5}$ and weight decay is $10^{-5}$. We train 6 epochs for 2D object detection and an additional epoch for 3D object representation learning. The object association module is learned for 4 additional epochs. The cosine learning rate scheduler is adopted. The warm-up stage is the first 500 iterations. In the label generation stage, threshold $\delta$ and $\kappa$ are 0.5 and 30.

**Inference.** We set the detection threshold to 0.5 and the bounding box whose score is below the threshold will not create a new tracklet. Similar to ByteTrack (Zhang et al., 2022b), the detections with lower scores are matched to the tracklet after the high-quality detections. We set the appearance threshold to 0.6 in data association, and we do not match the pair whose appearance similarity is below the threshold. Like DeepSORT (Wojke et al., 2017), we add an additional 2D IoU-based matching for the remaining detections and tracklets. 3D similarity weight $\alpha$ in matching is set to 0.4. The time interval before we terminate a tracklet, called max-age, is 30 frames.

---

[1]https://github.com/cheind/py-motmetrics

|  | MOTA ↑ | IDF1 ↑ | FP ↓ | FN ↓ | ID Sw. ↓ |
|---|---|---|---|---|---|
| Baseline | 51.0 | 62.3 | 8709 | 331056 | 9100 |
| +low-quality dets (Zhang et al., 2022b) | 53.4 | 64.3 | 13058 | 309653 | 9005 |
| +GNN | 56.5 | 66.5 | 20381 | 278752 | 10129 |
| +3D representation | 57.6 | 68.1 | 33587 | 258066 | 9920 |

Table 3: Ablation study of P3DTrack.

| Methods | MOTA ↑ | IDF1 ↑ | FP ↓ | FN ↓ | ID Sw. ↓ |
|---|---|---|---|---|---|
| 2D + 3D feature (Ours) | 57.6 | 68.1 | 33587 | 258066 | 9920 |
| 2D feature | 56.5 | 66.5 | 20381 | 278752 | 10129 |
| 2D feature + 2D motion | 55.5 | 58.6 | 29586 | 263370 | 23979 |
| 2D feature + 3D motion | 57.3 | 63.3 | 33048 | 257936 | 13026 |

Table 4: Different 2D and 3D representations in P3DTrack.

## 4.3 COMPARISONS WITH STATE-OF-THE-ART METHODS

In Table 1, we show the results compared with SOTA methods on the WOD val set. QDTrack is the previous SOTA method with a FasterRCNN-based detector and the quasi-dense matching learning module. We outperform QDTrack by 0.3 MOTA with a worse DLA-34 backbone. The strict matching strategy and thresholds make the number of FP much lower than QDTrack. The CenterNet-based detector has a lower recall than the FasterRCNN detector in QDTrack, so the FN is much higher than QDTrack and our IDF1 is also suffering from the lower ID Recall problem. As shown in Table 2, we achieve new state-of-the-art performance on KITTI dataset compared with the methods without any additional human annotations and pre-trained models on the autonomous driving datasets. Especially for the association metrics, we improve AssA by 3.0 and have the lowest number of ID Switches. Because we do not train our model on other autonomous driving datasets, the detection performance is not the best and we do not outperform the methods with a pre-trained model on the PD dataset. However, compared with the monocular 3D MOT methods that use the additional 3D ground truth, we still outperform them.

## 4.4 ABLATION STUDY AND FURTHER DISCUSSIONS

**Ablation study.** We ablate each component in P3DTrack, shown in Table 3. Our baseline is modified DeepSORT (Wojke et al., 2017), but without delayed tracklet initialization. Considering the low-quality detections like ByteTrack (Zhang et al., 2022b), the MOTA increases by 2.4. With association learning with GNN and jointly learning 3D representation, MOTA and IDF1 metrics can improve by 4.2 and 3.8.

**Influence of 3D representation.** In P3DTrack, we consider the appearance feature similarity and 3D representation similarity and weight them with $\alpha$. As shown in Table 4, we conduct experiments to verify this kind of design is better than the method only with appearance features and with 2D Kalman Filter and 3D Kalman Filter motion models. Compared with appearance features only, we have 1.1 MOTA and 1.6 IDF1 improvement. If we use 2D KF to model the motion of bounding boxes, the performance is worse than with appearance features only. That is because the 2D motion cannot be modeled as a linear system, especially for the cars parking on the roadside. The 3D KF is a better motion model. However, without jointly learned 3D representation, the association is still not well enough, especially for the IDF1 metric.

| 3D representation | MOTA ↑ | IDF1 ↑ | FP ↓ | FN ↓ | ID Sw. ↓ |
|---|---|---|---|---|---|
| P3DTrack (Ours) | 57.6 | 68.1 | 33587 | 258066 | 9920 |
| SfM (Schönberger & Frahm, 2016) | 55.5 | 66.8 | 55823 | 249935 | 11145 |
| MiDaS v3 (Ranftl et al., 2022) | 55.7 | 63.4 | 34514 | 259471 | 21058 |

Table 5: Comparisons with different 3D representation.

| det thres | app thres | MOTA ↑ | IDF1 ↑ | FP ↓ | FN ↓ | ID Sw. ↓ |
|---|---|---|---|---|---|---|
| 0.4 | 0.6 | 51.4 | 64.9 | 117703 | 210305 | 17933 |
| 0.5 | 0.6 | 57.6 | 68.1 | 33587 | 258066 | 9920 |
| 0.6 | 0.6 | 50.1 | 63.0 | 12885 | 337208 | 4864 |
| 0.5 | 0.5 | 56.1 | 66.6 | 41097 | 257660 | 13359 |
| 0.5 | 0.7 | 58.0 | 66.9 | 28798 | 260921 | 9139 |

Table 6: Different detection and appearance thresholds.

| 3D weight $\alpha$ | MOTA ↑ | IDF1 ↑ | FP ↓ | FN ↓ | ID Sw. ↓ |
|---|---|---|---|---|---|
| 0.0 (2D feature) | 56.5 | 66.5 | 20381 | 278752 | 10129 |
| 0.4 | 57.6 | 68.1 | 33587 | 258066 | 9920 |
| 0.5 | 57.3 | 67.2 | 34091 | 258324 | 11733 |
| 0.6 | 56.8 | 66.0 | 34213 | 258583 | 14431 |
| 1.0 (3D feature) | 52.1 | 54.2 | 33366 | 265203 | 42443 |

Table 7: Different 3D representation similarity weight.

**Different 3D representation.** We also compare different 3D representation in Table 5. Compared with learning-free Structure-from-Motion 3D positions, our method has the generalization ability to predict the moving objects, but only with SfM the 3D position of the moving object is incorrect. Besides, the pre-trained depth model MiDaS v3 (Ranftl et al., 2022) can not estimate video-consistent depth for the object. Based on the relative depth, the 3D motion model can not predict the correct depth in the next frame, which is important to obtain a good 3D similarity.

**Influence of detection and appearance thresholds.** The detection threshold is to decide detection quality. And the appearance threshold is to avoid the mismatch between the detections and tracklets. Experiments in Table 6 show these thresholds are necessary for our method. Finally, we choose 0.5 and 0.6 for detection and appearance thresholds to balance the MOTA and IDF1 metrics.

**Influence of 3D similarity weight.** As mentioned in Sec. 3.3, we weight the 3D similarity and appearance similarity with $\alpha$. Table 7 shows the tracking results with different weights. When $\alpha = 0$, we only use the appearance feature to associate objects. Note that even only with the 3D representation learned from the video, not depending on any image features, the tracking results are not so bad, with 5.1 MOTA and 13.9 IDF1 decreases compared with the best performance. However, with the combination of appearance similarity and 3D representation similarity, performance is better.

**Additional cost compared with baseline.** Although we generate the pseudo 3D object labels and train an additional 3D object representation learning module, we do not add much extra computation overhead and memory cost. As shown in Table 8, the inference latency only increases by 12 ms, about 5.5% relatively slower than the baseline. The training memory and the number of network parameters also indicate the efficiency of our method.

| Methods | Backbone | Latency | Memory | #Param. |
|---|---|---|---|---|
| Baseline | DLA-34 | 219.8ms | 13.29GB | 22.65M |
| P3DTrack (Ours) | DLA-34 | 231.5ms | 13.33GB | 24.42M |

Table 8: Inference latency, training memory, and the number of parameters compared with the baseline model.

## 5 CONCLUSION

In this paper, we propose a new 2D MOT framework with the help of 3D object representation from the video itself, called P3DTrack. We design a pseudo 3D object label generation method from the reconstructed scene and 2D bounding boxes with identity. Supervised by the pseudo 3D object labels, the 3D object representation can be jointly learned with 2D object detection and object association. So, an online tracker can be easily achieved with the learnable association module. We conduct experiments on large-scale ego-centric datasets, KITTI and Waymo Open Dataset. Extensive experiments show the effectiveness of our method. Latency analysis also indicates that P3DTrack does not increase much computational overhead or memory usage.

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

## A    APPENDIX

In the supplementary materials, we describe the detailed network architectures and implementation details, perform additional experiments, and provide more analyses. The contents are as below:

- In Sec. B, C and D, we show the details of P3DTrack to help understand our method.
- In Sec. E, there are additional experiments on KITTI and analyses about our 3D object learning module.
- In Sec. F, visualization shows the success and failure cases.
- In Sec. G, we discuss the limitation and future work.
- In Sec. H, the reproducibility statement helps reproduce our work.

## B    3D PSEUDO LABEL GENERATION ALGORITHM

---

**Algorithm 1:** Generating pseudo 3D object labels from a video.

---

**Input:** video clip $\mathcal{V}$, camera intrinsic $\{\mathbf{K}_{c_k}\}$, camera pose $\{\mathbf{T}_t^k\}$, 2D bounding box with id $\{\mathbf{B}_j^t\}_{j=1}^{n_d}$

**Output:** pseudo 3D object label $\{\mathbf{o}_j^{t*}\}$

---

1  $\{\mathbf{P}_i^*\}_{i=1}^n \leftarrow \texttt{SfM}(\mathcal{V}, \{\mathbf{K}_{c_k}\}, \{\mathbf{T}_t^k\})$
2  $\mathcal{P} \leftarrow \varnothing$
3  // IntraPC for each object
4  **for** $j \in [1, n_d]$ **do**
5     **for** $t \in [1, T]$ **do**
6        $\mathcal{P}_j^t \leftarrow \texttt{IntraPC}(\{\mathbf{P}_i^*\}, \mathbf{B}_j^t)$
7        $\mathcal{P} \leftarrow \mathcal{P} \cup \mathcal{P}_j^t$
8  // InterPC for all foreground points $\mathcal{P}$ to generate $n_d'$ clusters
9  $\{\mathcal{P}_{j'}\}_{j'=1}^{n_d'} \leftarrow \texttt{InterPC}(\mathcal{P})$
10  // Assign ID to the point clusters
11  $\{\mathcal{P}_j^*\}_{j=1}^{n_d} \leftarrow \texttt{MatchID}(\{\mathcal{P}_{j'}\}, \{\mathbf{B}_j^t\})$
12  // Find barycenter for each cluster and transform to each camera frame
13  $\{\mathbf{o}_j^{t*}\} \leftarrow \texttt{FindCenter\&Transform}(\{\mathcal{P}_j^*\}, \{\mathbf{T}_t^k\})$
14  **return** $\{\mathbf{o}_j^{t*}\}$

---

## C    NETWORK ARCHITECTURE

In this section, we explain the detailed network design of P3DTrack. The parameters of the network architecture are shown in Table 9. The backbone is a standard 'dla-seg' architecture, please refer to (Zhou et al., 2019) for more details. The heads include classification head (Cls-head), 2D regression head (Reg-head), offset head (Offset-head), depth and depth uncertainty head (Depth-head & Uncer-head). The appearance model is the ResNet50 backbone followed by a global average

| Module | Keys | Values |
|---|---|---|
| DLA-34 | levels | 1, 1, 1, 2, 2, 1 |
| | channels | 16, 32, 64, 128, 256, 512 |
| DLA-up | channels | 64, 128, 256, 512 |
| | scales | 1, 2, 4, 8 |
| Cls-head-pre | Conv | 2* (3*3),64,256 |
| Reg-head-pre | Conv | 2* (3*3),64,256 |
| Cls-head | MLP | 256,3 |
| Reg-head | MLP | 256,4 |
| Offset-head | MLP | 256,2 |
| Depth-head | MLP | 256,1 |
| Uncer-head | MLP | 256,1 |
| App-model | backbone | ResNet50 |
| App-feature | channels | 512 |
| 3D-feature | channels | 512 |
| Obj-feature | fusion | concat&norm |
| GNN | #layer | 1 |

Table 9: Network architecture of P3DTrack

| Configuration | Values |
|---|---|
| optimizer | Adam |
| optimizer momentum | $\beta_1 = 0.9, \beta_2 = 0.999$ |
| weight decay | $1 \times 10^{-5}$ |
| learning rate | $8 \times 10^{-5}$ |
| learning rate scheduler | cosine decay |
| warmup iterations | 500 |
| epochs | 6+1+4 |
| augmentation | random flip |
| aug. probability | 0.5 |
| batch size | 8 |
| gradient clip | 5 |
| image size | 1920 $\times$1280 (zero padding) |
| loss name | heatmap loss, bbox loss, offset loss depth loss, reid loss, association loss |
| loss weight | 1.0, 1.0, 0.2, 0.1, NA (pretrained), 0.5 |
| sync BN | True |

Table 10: Training config of P3DTrack.

pooling layer and a fully connected layer with 512 channels. The 3D object feature is 512 channels predicted by a fully-connected layer from the 3D object position regression. The object feature is fused by concatenating the appearance feature and 3D feature and normalizing it. The object feature is aggregated by 1 layer GNN.

## D TRAINING DETAILS

In Table 10, we list the training config of P3DTrack. In P3DTrack, 2D object detection is first learned for 6 epochs and 3D object representation is learned for 1 epochs. Then 3D object representation and object association can be jointly learned for 4 epochs. The appearance model is pre-trained and frozen in the later training stage.

| | $\delta < 1.25 \uparrow$ | $\delta < 1.25^2 \uparrow$ | $\delta < 1.25^3 \uparrow$ | Abs Rel$\downarrow$ | Sq Rel$\downarrow$ | RMSE$\downarrow$ | RMSE log$\downarrow$ |
|---|---|---|---|---|---|---|---|
| SfM init. | 0.945 | 0.962 | 0.967 | 0.112 | 1.183 | 4.744 | 0.203 |
| P3DTrack (Ours) | 0.974 | 0.983 | 0.985 | 0.049 | 0.456 | 3.372 | 0.110 |
| training w/ 3D GT | 0.991 | 0.994 | 0.995 | 0.025 | 0.142 | 1.999 | 0.059 |

Table 11: 3D object depth accuracy analyses on KITTI *training* set. We ignore the predictions with depth>75m.

| | $\delta < 1.25 \uparrow$ | $\delta < 1.25^2 \uparrow$ | $\delta < 1.25^3 \uparrow$ | Abs Rel$\downarrow$ | Sq Rel$\downarrow$ | RMSE$\downarrow$ | RMSE log$\downarrow$ |
|---|---|---|---|---|---|---|---|
| SfM init. | 0.981 | 0.988 | 0.990 | 0.106 | 0.831 | 3.973 | 0.473 |
| P3DTrack (Ours) | 0.987 | 0.992 | 0.993 | 0.084 | 0.653 | 2.940 | 0.145 |
| training w/ 3D GT | 0.996 | 0.998 | 0.998 | 0.022 | 0.233 | 1.569 | 0.061 |

Table 12: 3D object depth accuracy analyses on WOD *training* set. We ignore the predictions with depth>75m. Note that 3D GT is annotated on LiDAR, so some objects do not have the 3D label. True depth error is lower than the values in table.

# E    ADDITIONAL EXPERIMENTS AND ANALYSES

## E.1    3D OBJECT LOCATION ACCURACY ANALYSES

In Table 11 and Table 12, we discuss the object depth accuracy of our P3DTrack on KITTI and WOD. The metrics are from the general depth estimation metrics (Eigen et al., 2014). Note that we only calculate the depth accuracy of the object centers. We can find that our P3DTrack, which trains 3D object representation from SfM initialization, has more accurate object depth estimation than the initialized pseudo labels. Note that we only consider the objects that have been assigned 3D pseudo labels when evaluating the SfM initialization. However, for P3DTrack, we evaluate depth of all detected objects, including the generalized objects without initial 3D pseudo labels. Even though, P3DTrack still outperforms the SfM initialization in object 3D location on both KITTI and WOD datasets.

## E.2    GAP BETWEEN OURS AND USING 3D GT LABELS

Table 11 and Table 12 also show the 3D object location accuracy gap between ours and the model trained with ground truth 3D object labels. We find that learning 3D object representation from SfM-initialized pseudo labels is a little worse than learning from GT 3D labels. However, the gap in 3D location accuracy is relatively small, which means our pseudo labels are practical for estimating the object's 3D location. The ablation study results in the main paper also show that 3D object representation can help object association. In Table 2, the comparison with monocular 3D MOT, QD-3DT and Mono3DT, shows our advantages.

## E.3    DYNAMIC OBJECT 3D ESTIMATION ANALYSES

As mentioned in Sec. 3.1 and 3.2 of the main paper, we can not assign labels for some objects because they are filtered in pseudo 3D object label generation stage, especially for the dynamic objects that have different moving directions from the camera. The 3D location of these objects can only be learned by our 3DRL module. In Table 13, we validate the performance of generalized 3D location learning for dynamic objects. Most objects are not generated 3D object labels and only 43.6% of the objects (static objects) have pseudo labels. The small gap between dynamic and static objects in results shows the generalization of our 3DRL module to the unlabeled dynamic objects.

## E.4    TRACKING RESULTS OF DIFFERENT CAMERAS

We report the tracking results conditioned on the camera in Table 14. It is obvious that the main FRONT camera has the best performance. The FRONT LEFT and FRONT RIGHT camera has a relatively well performance on both MOTA and IDF1. However, the results of side cameras are not as good as front cameras, because of the rapid movement of the camera relative to the objects and

| | ratio | $\delta < 1.25$ ↑ | $\delta < 1.25^2$ ↑ | $\delta < 1.25^3$ ↑ | Abs Rel↓ | Sq Rel↓ | RMSE↓ | RMSE log↓ |
|---|---|---|---|---|---|---|---|---|
| Dynamic | 56.4% | 0.985 | 0.990 | 0.992 | 0.087 | 0.922 | 3.349 | 0.152 |
| Static | 43.6% | 0.988 | 0.994 | 0.995 | 0.081 | 0.305 | 2.305 | 0.135 |
| All | 100% | 0.987 | 0.992 | 0.993 | 0.084 | 0.653 | 2.940 | 0.145 |

Table 13: Dynamic/static objects' 3D location performance on WOD.

| | MOTA ↑ | IDF1 ↑ | FP ↓ | FN ↓ | ID Sw. ↓ |
|---|---|---|---|---|---|
| FRONT | 57.6 | 68.1 | 33587 | 258066 | 9920 |
| FRONT LEFT | 58.5 | 66.6 | 11782 | 105092 | 6874 |
| FRONT RIGHT | 55.6 | 64.1 | 8566 | 79516 | 4765 |
| SIDE LEFT | 52.8 | 61.5 | 12741 | 112641 | 8506 |
| SIDE RIGHT | 49.4 | 59.1 | 8126 | 83075 | 7413 |

Table 14: Tracking results for each camera on WOD *val* set.

the crowded vehicles in parking lots that are often on the sides. To keep the generalizability and scalability of the algorithm, we do not set specific parameters for the side cameras.

# F  QUALITATIVE RESULTS

We show the qualitative results of baseline and P3DTrack in Fig. 4. When the object is heavily occluded, with 3D representation, P3DTrack can keep the tracklet and avoid mismatching. 3D representation is also useful for similar appearances, the new object will not match the existing tracklet, due to different 3D locations.

Besides, we provide videos to show the tracking results of P3DTrack on WOD and KITTI datasets. The sequences are 'segment-7650923902987369309_2380_000_2400_000' in WOD val set and '0011' in KITTI test set. Please refer to the attachments in the compressed zip file.

# G  LIMITATIONS AND FUTURE WORK

In this paper, we estimate the object's 3D location and represent it as a 3D feature. However, because the monocular depth estimation is an ill-posed problem, the learned object 3D location is not so good, especially for the object far away (>100m). We will try to learn 3D object representation from multi-frames like recent work (He et al., 2023; Li et al., 2022a; Wang et al., 2022a).

# H  REPRODUCIBILITY STATEMENT

We will release the training and inference codes to help reproduce our work and the documents will be clearly written. Pseudo 3D object labeling tools that are based on open-source packages

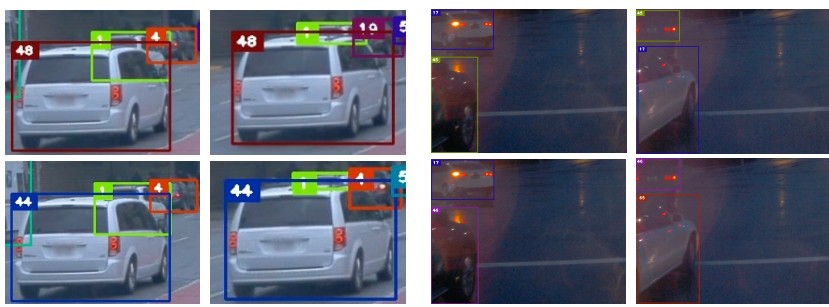

(a) Heavy occlusion.      (b) Similar appearance.

Figure 4: Qualitative results. Top: Baseline model without 3D object representation. Bottom: P3DTrack (Ours).

COLMAP (Schönberger & Frahm, 2016)[2] and hloc (Sarlin et al., 2019)[3] will be released. The checkpoint trained on the KITTI dataset will also be released. Limited by the license of WOD, the checkpoint of the model trained on WOD cannot be publicly available. However, we will provide it by email if needed. The implementation details, including the implementation details, network architecture, and different settings on KITTI are mentioned in Sec. 3 and 4.2 in the main paper and Sec. C, D and E in the appendix. The best hyperparameters of the experiments are listed in Sec. 4.2 in the main paper and Sec. E in Appendix, and the influences of hyperparameters are shown in Table 6 and Table 7. The inference latency and computational cost are in Table 8 in the main paper. The data and annotations of WOD[4] and KITTI[5] are publicly available.

---

[2] https://github.com/colmap/colmap

[3] https://github.com/cvg/Hierarchical-Localization

[4] https://waymo.com/open/

[5] https://www.cvlibs.net/datasets/kitti/eval_tracking.php

