# OpenReview forum: "Learning Pseudo 3D Representation for Ego-centric 2D Multiple Object Tracking"
_ICLR.cc/2024/Conference — Submitted to ICLR 2024_

### Official Review · Reviewer_yTKp · 2023-10-26

**Soundness:** 3 good
**Presentation:** 4 excellent
**Contribution:** 3 good
**Rating:** 8
**Confidence:** 4

**Summary:**

This work addresses the difficult problem of data mapping in 2D multiple object tracking (MOT), particularly in the context of object occlusions. While this problem is complicated in 2D, it is much easier to handle in 3D space using a 3D Kalman filter. The authors propose a new approach that uses 3D object representations to improve data mapping in 2D MOT.

In their method, referred to as P3DTrack, they use 3D object representations learned from monocular video data and monitored by 2D tracking labels, eliminating the need for manual annotation from LiDAR or pre-trained depth estimators. This approach differs from existing depth-based MOT methods in that it learns the 3D object representation along with the object association module.

The authors conduct extensive experiments and demonstrate the effectiveness of their approach by achieving the best performance on popular egocentric datasets such as KITTI and Waymo Open Dataset (WOD). They also commit to publishing the code for their method to make it accessible for further research and practical implementation.

**Strengths:**

+The study presents a novel approach to the task
+The proposed method is based only on RGB 2D input, which has the advantage that the hardware required is very simple and basic.
+Methods are evaluated on two public datasets. Choice of dataset is motivated and explained
+The paper is well structured and written. Study is well motivated
+Clear description of implementation and methods allows reproduction of experiments
+Authors perform ablation study with various experiments showing advantages of proposed architecture over other SOTA methods.

**Weaknesses:**

-Not clear in which dataset the ablation was performed? Is it for both or just one? It should be done for both datasets
-The paper lacks qualitative results. Instead, there is a figure in the supplementary material that clearly explains the problem and the improvement. The paper would benefit from more qualitative results like this. This could be done as a teaser figure on the first page, giving the reader a good overview of the topic and the contribution.
-It is not stated on which device the inference times are measured. Is it on the same GPU on which it was trained? One GPU or more? How far is it from working in real time? It is not clear and SLAM related applications require working in real time.

**Questions:**

1. Why do authors use the specified object detection method instead of something newer like DETR or YOLOv8?
2. How does the system work at night when visibility is reduced? Does the ability of 3D reconstruction remain the same or does it decrease?
The last "-" could be placed as question too.

---

> ### Author Response · Authors · 2023-11-22
>
> We sincerely thank you for the professional comments. We are inspired and hope our discussion brings more insights.
> ### W1: Ablation study on KITTI
> - We conducted ablation study on Waymo open dataset, because it is a more large-scale dataset than KITTI. Besides, there is no validation set on KITTI. The previous works that conducted ablation study on KITTI split each training sequence into two halves, and use the first half frames for training
> and the second for validation. The overlapping videos cause inaccurate (too high) results that cannot reflect the actual performance well.
> - We also supplement the ablation study on KITTI.
>
> |Method|HOTA↑|MOTA↑| FP↓| FN↓ |IDSW↓|
> |-|:-:|:-:|:-:|:-:|:-:|
> |Baseline| 77.6| 86.2| 6.3%|7.4%| 0.1%|
> |+low-quality dets| 78.1|86.5| 7.5%|5.9%| 0.0%|
> |+GNN |78.3 |86.5| 7.6%| 5.8%| 0.0%|
> |+3D representation|78.6 |86.8| 7.4%| 5.8%| 0.0%|
> ### W2: More qualitative results
> We show the qualitative results (videos with tracked objects) in supplementary material zip file. To show more results, we also build a project page https://p3dtrack.github.io/.
> ### W3: Inference time
> - We measure the latency on NVIDIA A40 GPU, the same as it was trained on. We use a single GPU to measure the latency.
> - The inference latency is mainly dominated by the object detector, for about 150ms per frame. The tracking stage (feature extraction and object association) takes about 80 ms. If using a real-time object detector, like the speed-optimized YOLO-series detector (YOLOX_x 17.3 ms, YOLOv8x 3.53ms), we believe it will be a (near) real-time system for autonomous driving (sensor input is 10Hz for most autonomous driving systems). Besides, please note the 3D reconstruction is only conducted in pseudo-label generation stage, and it will not affect the inference speed.
> ### Q1: Object detector
> CenterNet is a commonly used detector in MOT, so we chose it as our base detector for a fair comparison with SOTA methods. (Because the detector is not the core concern in our method.)
>
> We also conduct additional experiments on a more modern detector to show the performance with a better detector. We show the results with the YOLOX detector (also widely used recently, especially for ByteTrack-series trackers.)
> The detection performance is very well on Waymo open dataset.
> ||AP VEHICLE |AP PEDESTRIAN |AP CYCLIST|AP ALL|
> |-|:-:|:-:|:-:|:-:|
> |YOLOX|64.22|78.36|57.91|66.83|
>
> Based on it, we can achieve better tracking results.
> |Method| Backbone| MOTA ↑ |IDF1 ↑| FN ↓ |FP ↓ |ID Sw. ↓ |MT ↑ |ML ↓|
> |-|:-:|:-:|:-:|:-:|:-:|:-:|:-:|:-:|
> |QDTrack (Fischer et al., 2022)| ResNet-50 |55.6 |66.2 |514548 |214998 |24309 |17595 |5559|
> |P3DTrack (Ours)| DLA-34 |55.9 |65.6| 638390| 74802| 37478| 13643 |8583|
> |P3DTrack_YOLOX (Ours)| Modified CSP v5|59.2|67.6|549228|90342|51913|16475| 7362|
>
>
> ### Q2: At night
> When the visibility is low at night, the SfM system will be negatively affected.  However, it only slightly affects pseudo label generation. The network learning 2D boxes and 3D representation is less affected. We show a video on the anonymous project page.

---

### Official Review · Reviewer_e7hh · 2023-11-03

**Soundness:** 4 excellent
**Presentation:** 3 good
**Contribution:** 3 good
**Rating:** 6
**Confidence:** 4

**Summary:**

The paper proposes to jointly learn 3D features and their tracking association on top of existing 2D detector.
The supervision is provided via 3D MVS reconstruction of the static scene from moving camera, which allows extraction of 3D object locations and associate them with corresponding views.
This is shown to marginally improve MOTA accuracy on standard datasets.

**Strengths:**

The paper is well structured and readable.
The review of state of the art appears complete and up to date.
The central idea of joint association via auxiliary depth loss  and its learning approach are novel. The representation itself uses standard building blocks, but the specific architecture is of interest for the computer vision community.
Ablation study is included to validate parameter and component choices.
Implementation details appear sufficient for replication of results.

**Weaknesses:**

Quantitative results constitute only incremental improvements over SOTA.
There is no analysis of failure cases, especially w.r.t non-static classes, which can be missed in the pseudo GT association due to rigid scene assumption.

**Questions:**

Provide class based results, i.e. for pedestrians and cars to show there is no significant static bias.

---

> ### Author Response · Authors · 2023-11-22
>
> We are grateful for your valuable comments and constructive advice, which help us a lot to make the paper better.
> ### incremental improvements over SOTA
> The main reason is that we use the weak CenterNet-based object detector with DLA-34 backbone, even worse than FasterRCNN_R50 in QDTrack. On COCO, there is a 1~2AP detection performance gap (39.2 vs 37.4). (We use CenterNet because a single-stage object detector makes it easier to add a 3D representation head.) However, we also show the performance increase with object detector scaling-up. When using a better detector YOLOX to compensate for the detector performance gap, we can outperformance SOTA methods by a large margin.
> |Method| Backbone| MOTA ↑ |IDF1 ↑| FN ↓ |FP ↓ |ID Sw. ↓ |MT ↑ |ML ↓|
> |-|:-:|:-:|:-:|:-:|:-:|:-:|:-:|:-:|
> |QDTrack (Fischer et al., 2022)| ResNet-50 |55.6 |66.2 |514548 |214998 |24309 |17595 |5559|
> |P3DTrack (Ours)| DLA-34 |55.9 |65.6| 638390| 74802| 37478| 13643 |8583|
> |P3DTrack_YOLOX (Ours)| Modified CSP v5|59.2|67.6|549228|90342|51913|16475| 7362|
>
> ### Failure cases
> The common failure case of our method is the occluded scene, like parking lot, distant objects, occluded objects very close to the camera. The 3D location for these cases in naturally hard to estimate.
> The detailed qualitative results can be found on the anonymous project page: https://p3dtrack.github.io/.
>
> ### Non-static object
> Firstly, we show the object-level depth estimation results between car and pedestrian.
> |Category|$\delta< 1.25\uparrow$  |$\delta< 1.25^2\uparrow$|$\delta < 1.25^3\uparrow$  |Abs Rel$\downarrow$ |Sq Rel$\downarrow$ |RMSE$\downarrow$ |RMSE log$\downarrow$ |
> |-|:-:|:-:|:-:|:-:|:-:|:-:|:-:|
> |Car|0.993|0.995|0.996|0.093|0.558|3.367|0.198|
> |Pedestrian|0.981|0.992|0.994|0.055|0.292|3.086|0.104|
>
> The results show that even though pedestrians are more likely to be dynamic, the results for pedestrian are still no worse than vehicle.
> The object tracking results for these two classes are shown in the following Table.
>
> |Category| MOTA ↑ |IDF1 ↑| FN ↓ |FP ↓ |ID Sw. ↓ |MT ↑ |ML ↓|
> |-|:-:|:-:|:-:|:-:|:-:|:-:|:-:|
> |QDTrack_Car| 55.6 |66.2 |514548 |214998 |24309 |17595 |5559|
> |QDTrack_Pedestrian| 50.3| 58.4|151957 |48636| 6347| 3746 |1866 |
> |Ours_Car| 59.2|67.6|549228|90342|51913|16475| 7362|
> |Ours_Pedestrian|55.7| 57.8|139181|28307 |16829 | 4029|1761 |
>
>
> Then we discuss more about why there is no significant static bias.
> Our method for 3D representation learning contains two main parts: 3D pseudo-label generation and 3DRL with 3D pseudo label.
> - As for 3D pseudo-label generation, "the points on dynamic objects are mostly ignored". That means 3D pseudo labels are rarely generated for pedestrians due to their movement. (However, when they are static, such as waiting for a red light, the pseudo label can still be generated.) This is the natural shortage of SfM.
> - Although lacking 3D pseudo labels, we can utilize the "generalization ability of depth from other objects" of the object 3D representation learning module. The other objects are mainly vehicles that are learned with 3D pseudo labels. We can generalize the depth of pedestrians from the learned depth of vehicles. This generalization ability is because depth is the class-agnostic attribute of the object, and the network learns depth from the whole image. During the inference stage, the network can leverage the learned depth information of other vehicles in the same image to predict the depth of pedestrians.

---

### Official Review · Reviewer_RXkh · 2023-11-03

**Soundness:** 2 fair
**Presentation:** 2 fair
**Contribution:** 1 poor
**Rating:** 5
**Confidence:** 5

**Summary:**

A 2D multiple object tracker is proposed that consists of 3 steps, an object detector, a 3d descriptor and then an associator consisting of matching 3d features.   The paper provides good results on datasets from KITTI and Waymo and compares to other tracking algorithms.

**Strengths:**

The paper provides results on 2 driving datasets.  There is a 3 stage process, the first is acquiring depth values using SFM which is an offline process.  The 2nd stage uses a MLP to derive a 3d representation which is based on clusters of 3d points.  The third stage is the data association.  The results are good, slightly better than the other algorithms compared to.

**Weaknesses:**

The paper does not really compare to state of the art methods for MOT.  There are only 2 driving datasets compared to, the WayMo and Kitty datasets.  Also, the datasets chosen do not really demonstrate the ability to do multiple object tracking as at most there is 2  or at most 3 objects being tracked.  With regards to MOT, the datasets chosen should have been from the MOT challenge https://motchallenge.net.  The authors claim that using 3d features simplifies the problem, which it does for the dataset used, as objects are clearly separated by depth.  In addition, datasets that should have been considered also include GMOT-40 and Animal Track.   Long term pixel tracking is of interest recently which makes sense when objects are not separated by depth as in the examples given, some papers to look at include Tap-vid, Tapir, Particle video revisited, and tracking everything everywhere all at once.
I do not see how this method can perform real time tracking at all, first you need to train on a dataset and then you use SFM Colmap which is an offline process.  No generalizations of the method are demonstrated.  It would be interesting to see how well the method works if it was trained on WayMo and tested on Kitty and visa versa.

**Questions:**

Is this more than just a simple tracking by detection?  You do not appear to do a predict and correct by detection or observation as would be necessitated by a kaman filter.
For all your examples, the objects are well separated by depth, what happens when they are not?

---

> ### Author Response · Authors · 2023-11-13
>
> Thanks for your comment. However, we kindly hope the reviewer can read our paper again, because some comments deviate from the facts.
>
> ### Response to the comments in Strengths:
> - Why talk about LiDAR-based 3D MOT and why cite Kalman Fiter in the introduction?
>
> We observed that using only the Kalman filter works very well for LiDAR-based 3D MOT. Therefore, we hope to help 2D MOT with a 3D representation.
> ### W1: The paper does not really compare to state of the art methods for MOT.
> We tried our best to compare the SOTA methods. As for KITTI dataset, please refer to https://www.cvlibs.net/datasets/kitti/eval_tracking.php for the tracking leaderboard. All published image-based SOTA methods are compared in Table 2.
> For Waymo Open Dataset, we compared our method with real SOTA QDTrack. For other methods not reported on these two datasets, such as ByteTrack, we implemented it using the official code and reported it in our ablation study (Table 3 Row 2).
> We think we have sufficiently compared the existing SOTA MOT methods.
>
> ### W2: Also, the datasets chosen do not really demonstrate the ability to do multiple object tracking as at most there is 2 objects being tracked.
> It seems the reviewer is not familiar with these datasets. As the widely-used MOT datasets, KITTI and Waymo have far more than **2 objects** being tracked. More than 100 methods report MOT results on these datasets.
> We believe that these datasets can demonstrate the capability of MOT.
>
> ### W3: With regards to MOT, the datasets chosen should have been from the MOT challenge.
> As mentioned in Section 4.1, we discussed why we don't choose MOT17 and MOT20 as the evaluation dataset.
> >Please note that other datasets, such as MOT17 and MOT20, include numerous surveillance videos without camera motion. These datasets are not suitable for evaluating our method.
>
> SfM cannot be operated in non-geo-centric videos. Ego-centric video means the camera is moving along with the ego.
> ### W4: The authors claim that using 3d features simplifies the problem, which it does for the dataset used, as objects are clearly separated by depth.
> Please note that as a pure image-based MOT method, objects cannot be **clearly separated by depth**, how we separate them without any depth labels is indeed our contribution. Besides, pretrained depth cannot beat us. (Table 5)
>
> ### W5: In addition, datasets that should have been considered also include GMOT-40 and Animal Track.
> The GMOT-40 and Animal Track datasets have almost no necessary connection with our approach and our ego-centric setting. And compared with the datasets we used, GMOT-40 and Animal Track are much less popular.
>
> ### W6: Long term pixel tracking is of interest recently which makes sense when objects are not separated by depth as in the examples given.
> We appreciate your advice, but we don't know the necessity of using long term pixel tracking methods in our paper for now. Pixel tracking and object tracking are totally different tasks. We agree that it may help to track objects, however, there is no connection with our pseudo 3D representation.
> ### Q1: Is this more than just a simple tracking by detection?
> As shown in the introduction, we summarize the contribution of the paper.
> >(1) We propose a new online 2D MOT paradigm, called P3DTrack. P3DTrack utilizes the
> jointly learned 3D object representation and object association.
> (2) We design the 3D object representation learning module, called 3DRL. With 3DRL, the
> object’s 3D representation is learned from the video and supervised by the 2D tracking
> labels without any additional annotations from LiDAR or pretrained depth estimator.
> (3) The extensive experiments on the large-scale ego-centric datasets, KITTI and Waymo Open
> Dataset (WOD) show that we achieve new state-of-the-art performance.
>
> The most important contribution lies in the new pseudo 3D representation without depth/LiDAR supervision.
>
> ### Q2: You do not appear to do a predict and correct by detection or observation as would be necessitated by a kaman filter.
> As mentioned in Section 3.3, the proposed tracker infers **similar to DeepSORT**. In DeepSORT, the tracker predicts and corrects the tracklet by detection. Since we did not specifically mention, we don't change this process.
>
> ### Q3: IF you are using a Kalman filter, this is linear and will fail when motion is not linear?
> No. As discussed in many MOT papers, Kalman Filter can work for regular high-frame-rate videos.
>
> ### Q4: For all your examples, the objects are well separated by depth, what happens when they are not?
> As mentioned in Section 3.2, besides 3D representation, we also have appearance model just like other papers. When 3D representation fails, the appearance model can provide available association.
>
> **In summary, it appears that there are many misunderstandings in comments regarding our work and the field of MOT. Please consider reading related work and our paper again.**

---

> > ### Comment · Reviewer_RXkh · 2023-11-15
> >
> > W1. 2 data driving sets are used for testing, this is limiting to make general comments about applicability to MOT in general.
> > W2.  I am familiar with the datasets, they are 2 driving datasets.
> > W3. To make general MOT claims, there should be other datasets besides driving ones.
> > W4. You use SFM Colmap to get a depth estimate.  SFM only provides estimates to scale, this is not mentioned however since everything is to scale it would not matter.  I am not sure what your 3D representation adds, what if you just used the SFM clustered depth estimate as your representation, I would bet the results would be the same.
> > W5.  Why not?  The paper should be clear on what datasets this method works on, the authors hint that it is a general MOT solution so why not demonstrate this.
> > W6. Yes no relation to the pseudo 3d representation, however it should be referenced as another approach.
> > Q1.  the pseudo representation has not been demonstrated to add more than an average depth in your ablation studies or elsewhere.
> > Q2. Ok, same as DeepSort, I guess you use a graph matching.
> > Q3.  Ok, you are not using Kalman filter.  I did find the paper a bit difficult to follow and hard to read.
> > Q4.  See my previous comments.
> >
> > new question:  How does this method function in real time given SFM is an offline process,
> > new question: How does this method generalize?  What if we learned with Wayne and tested with Kitty and visa-versa?
> >
> > I have updated my ratings, perhaps I was a bit harsh at the beginning

---

> > > ### Author Response · Authors · 2023-11-23
> > >
> > > Thanks again for your comment. We think the reviewer still has some misunderstandings regarding our work.
> > >
> > > ### W1 & W2: dataset
> > >
> > > - There are more than 100k objects in the Waymo open dataset and about 1k objects in KITTI being tracked. "2 or at most 3 objects" being tracked deviate from the facts.
> > > - If the statement "2 or at most 3 objects" means the 2 or 3 *categories* instead of *objects*, it is still not a weakness. In MOT task, especially for tracking by detection paradigm, most methods focus on the association within the same category. Even if there are multiple categories, they still associate the detections separately. Besides, the MOT17 and MOT20 datasets also have *ONE* category (pedestrian).
> > > - About driving datasets. We focus on ego-centric scenes, and the autonomous driving scene is the most important for ego-centric/embodied AI to demonstrate the effectiveness of methods. Another scenario is indoor robotics, where there are few dynamic objects, so MOT is not highly regarded. Therefore, we choose the autonomous driving datasets.
> > >
> > > ### W3: MOT Challenge results
> > >
> > > We also report the additional results on general non-ego-centric dataset MOT17 test set. The results show that even though training 3D representation on only half of the videos that have ego-motion, we still obtain performance improvement, especially compared with pretrained depth-based tracker QuoVadis [D].
> > >
> > > | Method | MOTA↑ | IDF1↑ | HOTA↑ | FP↓ | FN↓ | IDSw↓ |
> > > | --- | --- | --- | --- | --- | --- | --- |
> > > | ByteTrack | 80.3 | 77.3 | 63.1 | 25491 | 83721 | 2196 |
> > > | QuoVadis | 80.3 | 77.7 | 63.1 | 25491 | 83721 | 2103 |
> > > | BoT-SORT | 80.5 | 80.2 | 65.0 | 22521 | 86037 | 1212 |
> > > | P3DTrack (Ours) | 80.6 | 79.8 | 64.9 | 22119 | 86061 | 1197 |
> > >
> > > ### W4: comparing with SfM depth
> > >
> > > In Table 5, we have compared the results of using only SfM for depth estimation (row 2). With our 3D representation, P3DTrack has better performance than SfM.
> > > We have also explained why the estimated depth is better than SfM estimation.
> > >
> > > > Compared with learning-free Structure-from-Motion 3D positions, our method has the generalization ability to predict the moving objects, but only with SfM the 3D position of the moving object is incorrect.
> > >
> > > Here, we explain how the generalization works in detail.
> > > Our method for 3D representation learning contains two main parts: 3D pseudo-label generation and 3DRL with 3D pseudo label.
> > >
> > > - As for 3D pseudo-label generation, "the points on dynamic objects are mostly ignored". That means 3D pseudo labels are rarely generated for moving objects due to their movement. (However, when they are static, such as waiting for a red light, the pseudo label can still be generated.) This is the natural shortage of SfM.
> > > - Although lacking 3D pseudo labels, we can utilize the "generalization ability of depth from other objects" of the object 3D representation learning module. The other objects are static objects that are learned with 3D pseudo labels. We can generalize the depth of moving objects from the learned depth of static objects. This generalization ability is because depth is the class-agnostic attribute of the object, and the network learns depth from a single image.
> > >
> > > Please consult the table in response to Reviewer e7hh's question about “Non-static object”.
> > >
> > > ### W5: other datasets
> > >
> > > We have supplemented the results on MOT17 dataset. As for these mentioned datasets, because they are not ego-centric videos and are not widely used datasets.
> > >
> > > ### W6: pixel tracking works
> > >
> > > We will consider citing them when we introduce the different tracking methods in the revised paper.
> > >
> > > ### New question 1: real-time
> > >
> > > Please note that SfM is only conducted in the training stage. During inference, we use the 3D representation from the output of the 3DRL network. So, SfM does not affect the inference time cost.
> > >
> > > ### New question 2: generalization between datasets
> > >
> > > The generalization ability across datasets are mainly dominated by object detector. When trained on Waymo open dataset (WOD) and tested on KITTI dataset, the performance will drop by about 4 AP (88.1 vs. 84.3) for vehicles. The performance does not decrease much because WOD is much larger than KITTI. And when trained on KITII and tested on WOD, the performance will drop a lot (more than 15 AP).

---

### Official Review · Reviewer_kQ8t · 2023-11-04

**Soundness:** 3 good
**Presentation:** 3 good
**Contribution:** 2 fair
**Rating:** 5
**Confidence:** 5

**Summary:**

The paper introduce a pipeline for solving 2D multiple object tracking by learning 3D object representation, 2D object appearance feature and an object association model.
The 3D object representation is learnt from Pseudo 3D object labels created from monocular videos using structure-from-motion approach.
The object association model consists of two components: GNN-based feature aggregation and a differentiable matching.
The experiments conducting on KITTI and Waymo Open Dataset demonstrate the effectiveness of the method.

**Strengths:**

1. Paper is well written with extensive experiments to support the proposed method.
2. A 2D MOT method that can leverage the power of 3D representation without LiDAR data or a depth estimation model.

**Weaknesses:**

1. The novelty of the paper is limited. The main idea is to generate pseudo 3D object labels from monocular videos so that it can be used to train the model to obtain 3D location / representation. There are main issues and details need to address about the process of generating these pseudo labels. Thus, in terms of the methodology for MOT, there is not much new development in this paper, e.g. GNN-based aggregation, association model learning and appearance using reID feature, etc. are the existing techniques in MOT literature.
2. The impact of the paper is limited given the ego-centric datasets with LiDAR data widely available.

**Questions:**

1. The author should provide more details on how to filter low speed of ego-motion videos and moving objects. It is also not clear how tracklet of those moving objects being handle.
2. How can the model learn if there is only loss to supervised static object? The output o^t_j can be any values.
3. I would like to see how is the quality of pseudo 3D object labels impact on the performance of learned 3D representation. One ablation study can be done is to use real 3D object labels to train the model and compare.
4. In table 3, there is an increase of ID Sw when using Baseline + 3D representation. How do you explain this behaviour? Is it suppose for 3D representation to help reduce ID Sw.

---

> ### Author Response · Authors · 2023-11-22
> **Response to Reviewer kQ8t (Part 1)**
>
> We really appreciate your valuable comments. We are trying our best to address your concerns and are open to any further discussions.
> ### W1: Limited novelty
> - Our main novelty and contribution lie in the 3D representation. Specifically, We focus on how to obtain 3D labels, how to generalize the label to other objects, and how to use the (not so accurate) 3D features in MOT. The experiments also show that our 3D representation yields a substantial improvement compared to traditional 2D representation and other pretrained 3D features. As for technical contribution, to obtain 3D labels, we design the two-stage object clustering (IntraPC and InterPC). To generalize the label to other objects, we propose an image-based 3DRL module learning from all 2D labels and static 3D labels, and verify that this learning module can generalize the static labels to dynamic objects. To measure the degree of inaccuracy, we introduce uncertainty-based 3D loss.
> We believe this learning paradigm can inspire the MOT community to pay attention to mining the 3D information hidden in the video that does not require manual annotation.
> - The feature aggregation and association modules are not the main contribution in our paper. For the online "Tracking by Detection" paradigm, most methods share a very similar association module, i.e., encoding the appearance/motion features, aggregating the object features, matching between detections and tracklets. This pipeline design can even back to DeepSORT published in 2017, and further adopted and slightly modified by the following work GMTracker [A], ByteTrack [B], BoT-SORT [C], etc. In this paper, we only utilize the best practices in the data association stage to obtain better performance and do not focus on the novel design in this part.
> - About the details in 3D pseudo label generation, we try our best to solve your concern in the following.
>
> ### W2: Limited impact
> - We agree that most **existing** ego-centric datasets have LiDAR sensors. However, with the development of embodied AI and vision-based autonomous driving, hand-held cameras and cameras on vehicles without LiDAR will be widely used. For example, using DJI drones, driving Tesla autonomous-driving vehicles, and biking with a GoPro camera will provide more available data.  We believe our algorithm will have great potential in these fields.
> - Besides, we also report the additional results on the general non-ego-centric dataset MOT17 test set. The results show that even though training 3D representation on only half of the videos that have ego-motion, we still obtain performance improvement, especially compared with pretrained depth-based tracker QuoVadis [D]. Please note in this Table, we change our base detector to YOLOX to align the detector performance with SOTA methods.
>
> |Method| MOTA↑| IDF1↑| HOTA↑| FP↓| FN↓| IDSw↓|
> |-|:-:|:-:|:-:|:-:|:-:|:-:|
> |ByteTrack| 80.3| 77.3| 63.1 |25491| 83721 |2196|
> |QuoVadis|80.3|77.7|63.1|25491|83721|2103|
> |BoT-SORT| 80.5	|80.2	|65.0	|	22521	|86037|1212|
> |P3DTrack (Ours)|80.6 |79.8 |64.9 |22119 |86061 |1197|
>
> [A] He et al. Learnable Graph Matching: Incorporating Graph Partitioning with Deep Feature Learning for Multiple Object Tracking. In CVPR 2021.
>
> [B] Zhang et al. ByteTrack: Multi-Object Tracking by Associating Every Detection Box. In ECCV 2022.
>
> [C] Aharon et al. BoT-SORT: Robust Associations Multi-Pedestrian Tracking. arXiv 2206.14651.
>
> [D] Dendorfer et al. Quo Vadis: Is Trajectory Forecasting the Key Towards Long-Term Multi-Object Tracking? In NeurIPS 2022.

---

> ### Author Response · Authors · 2023-11-22
> **Response to Reviewer kQ8t (Part 2)**
>
> ### Q1: Filtering
> - We filter low speed of ego-motion videos according to the ego speed from GPS/IMU. The videos in which the ego-vehicle moves less than 20m will not be considered in the reconstruction. In Waymo open dataset, this threshold can easily separate low-speed and normal videos.
> - We filter the moving objects by the threshold of reconstructed 3D points. Moving objects have fewer reconstructed 3D points because the movement is not consistent with ego and cannot be reconstructed with many 3D points. The threshold is 30 3D points. Typically, there are more than 30 keypoints extracted from the object in one frame. If this threshold is not met, it indicates that there aren't enough matched keypoints between two frames for this object, suggesting that it may be a moving object.
> - About how moving objects are being handled:
> (1) As for 3D pseudo-label generation, "the points on dynamic objects are mostly ignored". That means 3D pseudo labels are rarely generated for moving objects, including moving vehicles and most pedestrians. (However, when they are static, such as a vehicle/pedestrian waiting for a red light, the pseudo label can still be generated.) This is the natural shortage of SfM.
> (2) Although the 3D representation label is not generated for moving objects, we can generalize the 3D representation from the static objects. This is because the 3DRL module learns object's 3D representation from a **single image** instead of a video. In the image, network predicts the 3D representation from appearance instead of motion, so moving and static objects are not different. Even though we only learn from static objects, the object's depth can be **generalized** to other moving objects. In detail, we can utilize the "generalization ability of depth from static objects" of the object 3D representation learning module.  This generalization ability is because depth is the class-agnostic attribute of the object.
> ### Q2: Learning from static object
> Just as mentioned in response to Q1, the 3DRL module learns object's 3D representation from a single image. Like monocular 3D object detector, on the head of CenterNet, we regress the object's 3D position ($H\times W\times 3$) from $H\times W$ feature map. However, the supervision is only on the 2D center of static objects. As mentioned in response to Q1, we learn class-agnostic 3D representation from monocular image, and in an image, moving and static objects are not different in appearance. Moving and static objects share the same prediction head, so $o^t_j$ will not be any values for moving objects.
>
> ### Q3: Impact of pseudo label quality
> We conduct additional experiments using real 3D object labels to show the impact of pseudo label quality. Please note that, in Waymo open dataset, real 3D object labels are from LiDAR (0-75m, VFOV [-17.6, +2.4)), which means 3D labels are fewer than 2D labels. Besides, the 3D object ID GT and 2D object ID GT are not aligned (the same object has a 3D ID and a 2D ID). So, we match the 3D GT and 2D GT using the maximum IoU matching. We also retrain our model to fit the objects in this subset.
>
> |Method| MOTA ↑ |IDF1 ↑| FN ↓ |FP ↓ |ID Sw. ↓ |
> |-|:-:|:-:|:-:|:-:|:-:|
> |3D GT|51.4|66.7|69495 |92768 | 11765|
> |3D pseudo label|51.1|64.1| 69611|92884|12557|
> ### Q4: Increase of ID Sw
> - It is because in the last row, the object recall is much higher than the baseline. Please refer to the FP and FN metrics in Table 3.
> ID Switches can only be compared under similar recall performance. In general, the more objects considered in tracking, the more ID Switches are reported.
> Besides, IDF1 is also the metric to reflect the identification performance. We obtain 5.8 IDF1 improvement (68.1 vs. 62.3).
> - 3D representation can help reduce ID Sw. Please refer to Table 4. In this Table, we only change the 2D/3D representation without modifying other modules and thus maintain a similar recall, so it can reflect the influence of our 3D representation better.

---

### Meta-Review · Area_Chair_FeNj · 2023-12-09

**Metareview:**

The paper was reviewed by 4 experts and initially received mixed reviews (5, 1, 6, 8). The reviewers had the following concerns:

1. limited novelty - using existing MOT techniques [kQ8t]
2. limited impact [kQ8t]
3. how does the quality of the pseudo labels impact performance? [kQ8t]
4. missing details about filtering and how the model learns using only loss on static objects [kQ8t]
5. no comparison with MOT SOTA [RXkh]
6. empirical evidence is limited, only 2 driving datasets, and only 2-3 objects [RXkh]
7. not real time [RXkh]
8. unclear how well it generalizes from one dataset to another [RXkh]
9. unclear how well the method works if objects are not well separated by depth [RXkh]
10. incremental improvement over SOTA [e7hh]
11. no analysis of failure cases [e7hh]
12. missing class-based results [e7hh]
13. missing qualitative results [yTKp]
14. missing details (ablation dataset? GPU? runtime?) [yTKp]
15. why not use better detectors [yTKp]
16. does it fail during night (because the reconstruction ability fails}? [yTKp]

The authors wrote a response to address the concerns. During the discussion period, Rev RXkh was still concerned about the lack of experiments on other MOT datasets, given that the 2 driving datasets only have 2-3 objects tracked at a time (the authors confuse this to mean 2-3 objects in all videos, or 2-3 object classes). Specifically, it is unclear how the method performs when objects are not separated clearly by depth. Rev kQ8t shared similar concerns. Rev e7hh was still concerned about the incremental improvement over SOTA, even after YOLOX is added to the pipeline, as well as properly showing fair comparisons with the same backbones. Rev yTKp was satisifed with the rebuttal.  After the discussion, RXkh and kQ8t were still negative, while others were positive.

Overall, the AC agrees with the concerns about limited evaluation and applicability to other MOT datasets (since the paper is written as a general 2d MOT paper), as well as the incremental improvement and incremental novelty (the main contribution is on obtaining the 3d labels). Showing broader applicability and better improvement would better justify the proposed method. Thus, the AC recommends reject.

**Justification For Why Not Higher Score:**

- limited evaluation and applicability to other MOT datasets (since the paper is written as a general 2d MOT paper)
- incremental improvement and incremental novelty.

**Justification For Why Not Lower Score:**

n/a

---

### Decision · Program_Chairs · 2024-01-16

Reject